# Two Years and Four Time Points: Description of Emotional State and Coping Strategies of French University Students during the COVID-19 Pandemic

**DOI:** 10.3390/v14040782

**Published:** 2022-04-10

**Authors:** Elodie Charbonnier, Aurélie Goncalves, Cécile Puechlong, Lucile Montalescot, Sarah Le Vigouroux

**Affiliations:** UNIV. NIMES, APSY-V, CEDEX 1, F-30021 Nîmes, France; aurelie.goncalves@unimes.fr (A.G.); cecile.puechlong@gmail.com (C.P.); lucile.montalescot@unimes.fr (L.M.); sarah.le_vigouroux_nicolas@unimes.fr (S.L.V.)

**Keywords:** mental health, lockdown, coping

## Abstract

While it is now clear that the COVID-19 pandemic has had a major impact on the mental health of individuals, especially the most vulnerable ones such as students, we have very little knowledge about the long-term consequences. The objective of this study was to compare the mental health and coping of French university students during the different phases of the pandemic in the first 2 years. To this end, French university students were evaluated at four time points: during France’s first lockdown (April–May 2020; *n*_T1_ = 1357), the period after lockdown (June 2020; *n*_T2_ = 309), 1 year after the first lockdown, which was also a lockdown period (April–May 2021; *n*
_T1′_ = 2569); and 1 year after the end of the first unlock (June 2021; *n*_T2′_ = 1136). Anxiety and depressive symptoms, coping and concerns were measured. In order to compare scores between the lockdown and unlock periods within the same year, paired samples *t*-tests were performed. To compare scores between the 2 years for different participants, independent samples *t*-tests were conducted. Our results showed that maladaptive strategies, concerns and symptoms were higher during lockdown periods, compared with unlock periods. In addition, symptom levels were higher in the second year of the pandemic compared with the first one. These argue that the psychological effects of COVID-19 were exacerbated by lockdowns but also by time. This highlights the need for more attention to be paid to students’ mental health.

## 1. Introduction

The COVID-19 pandemic has had deleterious effects on the mental health of students in different countries around the world, including high levels of anxiety and depressive symptoms [1,2,3,4,5,6]. Similar results have been observed among French university students [7,8,9,10,11,12,13]. These results may be explained in part by the fact that individuals who were already experiencing high levels of psychological distress prior to the pandemic, such as students, were more likely to have their mental health damaged by the COVID-19 pandemic [14,15]. Indeed, even before the pandemic, university students had been identified as having poorer mental health than their nonstudent peers [16,17,18], with 53% of students reporting depressive feelings since they started university [19]. A recent systematic review showed a prevalence of 26.1% for depressive symptoms and 24.5% for anxiety among university students [20]. In France, 30% of French university students have had symptoms of depression [21], and 5.9% have had a major depressive episode [22]. In addition, 55% of French university students feel anxious [23], and have major concerns related to academic success, their professional future, time management, exams and financial difficulties [24].

Long before the COVID-19 pandemic, numerous studies had highlighted the role of coping strategies in understanding poor mental health [25,26,27]. Coping strategies are the cognitive and behavioral efforts undertaken by individuals to deal with stressful situations [28]. They are classically categorized as maladaptive or adaptive [29]. The COVID-19 pandemic, particularly the lockdowns, exposed students to new difficulties that challenged their coping strategies [6,30]. For example, during lockdown, university students used more maladaptive strategies (e.g., denial) than usual [10]. In addition, studies have found associations between the coping strategies used to deal with the stress of the COVID-19 pandemic and anxiety and depressive symptoms [8,10,31].

Studies have highlighted the long-term deleterious mental health consequences of previous pandemics [32,33,34]. This has led some authors to suggest that the psychological effects of COVID-19 may be long lasting [35,36,37,38]. However, to date, we mainly have data for the first year of the pandemic, with the exception of a few recent studies [39,40] that have reported increases in anxiety and depressive symptoms in the second year of the pandemic. 

Therefore, the objective of the present study was to compare the mental health (concerns, anxiety and depressive symptoms) and adjustment (coping strategies) of French university students during different phases (lockdowns and periods without lockdowns) of the COVID-19 pandemic in the first 2 years (in 2020 and 2021). We predicted that during lockdowns, compared with periods after lockdown, university students would have more anxiety and depressive symptoms, be more worried, and use more maladaptive strategies and fewer adaptive ones. In addition, we predicted that in the second year of the pandemic, compared with the first year, university students would have more anxiety and depressive symptoms, be more concerned, and use more maladaptive strategies and fewer adaptive ones.

## 2. Materials and Methods

### 2.1. Participants and Procedure

Data were collected anonymously at four time points, via an online survey: (1) during France’s first national lockdown (23 April–10 May 2020; *n*_T1_ = 1357; *M*_age_ = 21.22 years ± 4.64), during which all lessons were conducted online; (2) during the period after lockdown (9–26 June 2020; *n*_T2_ = 309; *M*_age_ = 22.24 years ± 5.93), with some evaluations taking place online, and some face-to-face; (3) 1 year after the first lockdown, which was also a lockdown period (23 April–10 May 2021; *n*_T1′_ = 2569; *M*_age_ = 21.45 years ± 4.19), during which all lessons were conducted online; and (4) 1 year after the first unlock (9–26 June 2021, *n*_T2′_ = 1136; *M*_age_ = 21.63 years ± 4.58), with some evaluations taking place online, and some face-to-face. Participants in T2 were participants from T1 who agreed to complete the questionnaire a second time. Similarly, participants in T2′ were participants from T1′ who agreed to complete the survey a second time. For all our measurement time points, our sample included large proportions of women (between 73.61% and 82.20%) and first-year students (between 24.27% and 37.73%). They came from different French universities (mainly in Lorraine, Nîmes and Cergy) and different study fields (mainly psychology, language and sport science). Participants’ mean ages were 21.2 years (*SD* = 4.6) at T1, 21.4 (*SD* = 4.2) at T2, 22.2 (*SD* = 5.9) at T1′, and 21.6 (*SD* = 4.6) at T2′. Participants’ characteristics at the four time points are set out in Table 1. It is important to note that sociodemographic variables (age, level of education) had little influence on our variables of interest. Only gender seemed to have a small effect on the emotional state and coping strategies of our participants (See Appendix A, Table A1). More precisely, women had higher levels of anxiety and depressive symptoms than men. Women also tended to use more of the following coping strategies than men: seeking emotional and instrumental support, emotion expression, self-blame, and denial. By contrast, men tended to use more acceptance and humor than women. However, these trends were not found at all measurement times.

A link to the survey was sent by e-mail to teachers in different universities, who forwarded it to their students. The link to the survey was also distributed via social media by the students who responded, thereby allowing students from other universities to respond. Participants agreed to participate in this study after reading a consent form. They were informed that their participation was voluntary, and they could withdraw at any time. All the procedures contributing to this work were undertaken in compliance with the ethical standards of the relevant national and institutional committees on human experimentation and with the 1975 Declaration of Helsinki, revised in 2008.

### 2.2. Measures

Anxiety and depressive symptoms were assessed using the French version of the Hospital Anxiety and Depression Scale [41]. This 14-item self-report questionnaire measures the intensity of both anxiety and depressive symptoms during the previous week (a score ≤ 7 means no symptoms, a score of 8–10 means possible symptoms, and a score ≥ 11 means probable symptoms). The distribution of our participants according to the different threshold scores is shown in Appendix B, Table A2.

Coping strategies were assessed using a French-language situational version of the Brief-COPE [29]. At Times 1 and 1′, participants were instructed to refer to a stressful situation related to the lockdown. At Times 2 and 2′, they were asked to refer to a stressful situation related to the COVID-19 pandemic. This self-report scale measures 14 coping strategies, which are divided into adaptive strategies (active coping, planning, instrumental support, use of emotional support, venting, positive reframing, humor, acceptance, and religion) and maladaptive strategies (behavioral disengagement, self-distraction, self-blame, denial, and substance use). Participants rated each of the 28 items on a 4-point scale ranging from *Never* to *Always*. 

Concerns were measured with three questions probing participants’ (1) level of concern about their own health with regard to the COVID-19 pandemic, (2) level of concern about their relatives’ health with regard to the COVID-19 pandemic, and (3) the extent to which they felt that the COVID-19 pandemic and lockdowns compromised their professional future. Participants rated each of the questions on a scale ranging from 0 (*Not concerned*) to 100 (*Very concerned*). 

### 2.3. Statistical Analysis

In order to compare scores between the lockdown and unlock periods within the same year, paired samples *t*-test were performed. To compare scores between our 2 years with different participants, independent samples *t*-tests were conducted. Effect sizes are expressed as Cohen’s *d*. Data were analyzed using JASP software (version 0.11.1; Amsterdam, The Netherlands). 

## 3. Results

In accordance with our first hypothesis, means comparisons (Table 2) showed that students had higher levels (*d* between 0.37 and 0.59) of anxiety and depressive symptoms during periods of lockdown (T1 and T1′) than during periods after lockdown (T2 and T2′). Comparisons of proportions (Appendix B, Table A2) showed that higher proportions of students exhibited possible anxiety and depressive symptoms during the two lockdown periods than during the two periods after lockdown. In addition, comparisons of means (Table 2) show that their concerns about the health of their relatives (*d* between 0.50 and 0.81), their own health (0.30 < *d* < 0.39), and their professional future (0.11 < *d* < 0.25) were greater during periods of lockdown (T1 and T1′) than during periods after lockdown (T2 and T2′). Finally, concerning coping strategies, our results indicated slightly (*d* < 0.20) lower use of adaptive coping strategies (e.g., active coping, instrumental support), and slightly (*d* < 0.30) higher use of maladaptive coping strategies (e.g., denial, behavioral disengagement) during the lockdown periods, compared with the periods after lockdown. 

In accordance with our second hypothesis, participants assessed in the second year of the pandemic (2021) had slightly higher levels (0.17 < *d* < 0.31) of anxiety and depressive symptoms than those interviewed in the first year (2020). More precisely, higher proportions of students (Appendix B, Table A2) presented possible or probable anxiety and depressive symptoms during the 2021 lockdown (T1′) than during the 2020 lockdown (T1). Comparisons of the two groups of students during the two post-lockdowns periods (T2 vs. T2′) highlighted slightly higher levels (*d* = −0.28) of concern about their relatives’ health in 2021 than in 2020, but no significant differences were observed regarding their concerns about their own health. Furthermore, they were slightly more (0.17 < *d* < 0.31) concerned about their professional future during the 2021 lockdown (T1′) than during the 2020 lockdown (T1), but no significant difference was observed between the post-lockdown periods in 2020 and 2021 (T2 and T2′). In addition, comparisons of the two groups of students during the two lockdowns (T1 vs. T1′) highlighted slightly lower levels of the use of coping strategies in 2021 than in 2020, particularly adaptive strategies such as acceptance *(d =* 0.39) and positive reframing (*d* = 0.28). By contrast, students tended to use slightly more instrumental support (*d* = −0.16), and emotional support (*d* = −0.28). In addition, they used slightly more maladaptive coping strategies such as self-blame (*d* = −0.38), behavioral disengagement (*d* = −0.29), self-distraction (*d* = −0.22), substance use (*d* = −0.13), and denial (*d* = −0.10). Finally, comparisons between the two groups of students during the two periods after lockdown (T2 vs. T2′) showed quite similar trends for adaptive strategies, but with smaller effect sizes. By contrast, regarding maladaptive strategies, only self-blame seemed to be used more at T2′ than at T2 (*d* = −0.29). 

## 4. Discussion

It is now widely acknowledged that the COVID-19 pandemic has had very deleterious effects on students’ mental health. However, little is known about how these effects changed over time, particularly during the second year of the pandemic. Therefore, the main objective of this research was to compare mental health and adjustment in university students in different lockdown and unlock phases during the first 2 years of the COVID-19 pandemic (2020 and 2021).

Regarding anxiety and depressive symptoms, our results revealed that students’ mental health was more severely impaired during lockdown periods, compared with unlock periods. This can be explained by isolation [6,36,42,43], given the importance of peers at this time of life [44]. Moreover, distance learning imposed during lockdowns caused students to confront many challenges, both technical [42,44,45] and human [46]. 

Concerning coping strategies, our results indicated that during periods of lockdown, students used fewer adapted strategies (e.g., active coping) and more maladaptive strategies (e.g., denial) to manage pandemic-related situations, compared with unlock periods. The isolating and restrictive pandemic context may have hindered the use of usual means of adaptation [31]. Indeed, the pandemic, and even more so the various lockdowns, exposed students to new and stressful situations over which they had only limited control. 

Regarding comparisons made between the 2 years, our results showed that students’ symptoms were more severe in the second year of the pandemic (2021) than in the first one (2020). This trajectory is consistent with that observed by [40]. It was due in part to persistent loneliness and feelings of social isolation [47]. This supports the hypothesis that the psychological effects of COVID-19 may be longlasting [35,36,38]. 

Finally, students questioned in 2021 employed fewer adaptive and more maladaptive strategies than those in 2020. These data suggest that the experience of the first year of the COVID-19 pandemic did not allow students to develop new resources to better cope with the second year, and the opposite may even have occurred. This can be explained by the repeated, uncontrollable, and uncertain nature of the stressful events related to the pandemic. Over time, people realized that there was nothing they could do directly to solve the problem, but they could simply follow the government’s rules (e.g., stay home), which were largely reinforced by information in the media [48]. This type of event tends to increase stress and lead to poor adjustment [49]. Given that poor coping is positively associated with poorer mental health [25,26,27], the higher anxiety and depression scores observed during the second year of the COVID-19 pandemic may in part be explained by greater use of maladaptive strategies by students.

The present results need to be interpreted with caution, for the following reasons. First, our sample was predominantly female, and although the initial size of our two samples was large, it was substantially smaller for the last two time points. Our results therefore need to be replicated with a larger and more representative population. Second, the participants evaluated in 2021 were different from those evaluated in 2020, making inter-year comparisons difficult. Further research is warranted to consolidate and generalize these results, preferably adopting a longitudinal design. 

To our knowledge, our study is the first to compare mental health and coping strategies in French university students during different phases of the pandemic (here, in the first 2 years of the COVID-19 pandemic). Our results indicate that the psychological effects of COVID-19 were exacerbated by lockdowns, as well as by time. Thus, university students, who are already known to be a vulnerable population, may be particularly at risk in 2022. It is therefore essential that universities’ preventive medicine departments provide adequate psychological support for students, by focusing on interventions that reduce their anxiety and depressive symptoms, and improve their stress adjustment strategies. According to [50], offering practical advice on coping strategies and stress management can help reduce the negative consequences of the COVID-19 pandemic. Furthermore, as periods of lockdown are the riskiest, it seems essential to be able to implement remote therapeutic strategies. To do so, online self-help interventions (e.g., web page, Facebook page, e-learning) seem promising tools. They have been identified as being particularly valuable for university students [51,52,53], and have proven to be effective for students during the pandemic [54].

## 5. Conclusions

Studies conducted during previous pandemics have shown that the psychological consequences of pandemics can last well beyond the peak, especially among the most vulnerable populations. However, to date, we know very little about the long-term consequences of the COVID-19 pandemic. Our study shows that students’ mental health was most severely affected during periods of lockdown and during the second year of the pandemic (2021). This supports the hypothesis that the psychological effects of COVID-19 may persist even after the pandemic and highlights the need to continue to deploy interventions (face-to-face and/or online) aimed at reducing students’ anxiety and depressive symptoms, as well as improving their stress adjustment strategies. 

## Figures and Tables

**Table 1 viruses-14-00782-t001:** Characteristics of survey respondents at the four time points.

	2020					2021				
	T1		T2		*χ* ^2^	T1′		T2′		*χ* ^2^
	*n*	(%)	*n*	(%)		*n*	(%)	*n*	(%)	
Total	1357		309			2569		1136		
Gender					7.87 *					6.69 *
Female	1049	(77.30)	254	(82.20)		1891	(73.61)	875	(77.02)	
Male	286	(21.08)	46	(14.89)		655	(25.50)	247	(21.74)	
Other	22	(1.62)	9	(2.91)		23	(0.90)	14	(1.23)	
University					5.48					9.55
Cergy	.	.	.	.		389	(15.14)	167	(14.70)	
Lorraine	384	(28.30)	68	(22.01)		347	(13.51)	169	(14.88)	
Montpellier	.	.	.	.		154	(5.99)	66	(5.81)	
Nîmes	587	(43.26)	143	(46.28)		399	(15.53)	154	(13.56)	
Paris	.	.	.	.		142	(5.53)	57	(5.02)	
Strasbourg	225	(16.58)	60	(19.42)		134	(5.22)	83	(7.31)	
Other	161	(11.86)	38	(12.30)		1004	(39.08)	440	(38.73)	
Education Level					27.29 ***					6.18
Undergraduate										
First year	512	(37.73)	75	(24.27)		729	(28.38)	305	(26.85)	
Second year	331	(24.39)	93	(30.10)		627	(24.41)	279	(24.56)	
Third year	336	(24.76)	80	(25.89)		527	(20.51)	223	(19.63)	
Master’s										
Fourth year	86	(6.34)	27	(8.74)		327	(12.73)	159	(14.00)	
Fifth year	75	(5.53)	25	(8.09)		196	(7.63)	94	(8.27)	
PhD	12	(0.88)	8	(2.59)		75	(2.92)	45	(3.96)	
Other	5	(0.37)	1	(0.32)		88	(3.43)	31	(2.73)	
Study field										
Art-Design	62	(4.58)	7	(2.27)	49.505 ***	44	(1.72)	20	(1.77)	30.355 *
Biology	81	(5.98)	14	(4.53)		138	(5.39)	57	(5.04)	
Communication-Journalism	22	(1.62)	3	(0.97)		36	(1.41)	17	(1.50)	
Law-Economics-Management	63	(4.65)	11	(3.56)		355	(13.86)	113	(9.98)	
Education	26	(1.92)	6	(1.94)		81	(3.16)	45	(3.98)	
History	94	(6.94)	16	(5.18)		68	(2.65)	35	(3.09)	
Languages	146	(10.78)	17	(5.50)		170	(6.64)	73	(6.45)	
Literature	38	(2.81)	7	(2.27)		48	(1.87)	26	(2.30)	
Linguistics	24	(1.77)	6	(1.94)		20	(0.78)	8	(0.71)	
Math-IT-Physics-Chemistry	13	(0.96)	1	(0.32)		267	(10.42)	99	(8.75)	
Psychology	485	(35.82)	174	(56.31)		573	(22.37)	324	(28.62)	
Health	2	(0.15)	0	(0.00)		61	(2.38)	28	(2.47)	
Sociology-Philosophy	50	(3.69)	8	(2.59)		50	(1.95)	23	(2.03)	
Sport science	210	(15.51)	36	(11.65)		358	(13.97)	133	(11.75)	
Geography-Town planning	8	(0.59)	1	(0.32)		79	(3.08)	37	(3.27)	
Other	30	(2.22)	2	(0.65)		214	(8.35)	94	(8.30)	

Note. * *p* < 0.05. *** *p* < 0.001. T1: France’s first national lockdown (23 April–10 May 2020), T2: first period after lockdown (9–26 June 2020), T1’: 1 year after first lockdown, which was also a lockdown period (23 April–10 May 2021); T2’: 1 year after first unlock.

**Table 2 viruses-14-00782-t002:** Means (standard deviations) at four time points and mean comparisons.

	2020						2021						T1-T1′		T2-T2′	
	T1		T2		*t*	*d*	T1′		T2′		*t*	*d*				
	*M*	*SD*	*M*	*SD*			*M*	*SD*	*M*	*SD*			*t*	*d*	*t*	*d*
Concern about professional future	56.39	(29.81)	52.56	(29.93)	1.97 *	0.11	63.97	(27.86)	55.61	(28.51)	8.41 ***	0.25	−7.17 ***	−0.27	−1.66	−0.11
Concern about own health	32.14	(28.00)	22.92	(23.22)	5.33 ***	0.30	33.36	(28.20)	24.67	(22.78)	13.29 ***	0.39	−1.29	−0.04	−1.19	−0.08
Concern about relatives’ health	68.00	(27.42)	45.66	(28.52)	14.24 ***	0.81	65.97	(27.52)	53.65	(28.11)	16.97 ***	0.50	2.20 *	0.07	−4.42 ***	−0.28
Anxiety symptoms	8.61	(4.54)	7.15	(4.13)	6.47 ***	0.37	10.05	(4.65)	7.94	(4.38)	17.86 ***	0.53	−9.31 ***	−0.31	−2.86 **	−0.18
Depressive symptoms	7.03	(4.08)	4.49	(3.53)	10.44 ***	0.59	7.75	(4.30)	5.40	(3.94)	19.39 ***	0.58	−5.06 ***	−0.17	−3.67 ***	−0.24
Coping strategies																
Acceptance	6.14	(1.65)	6.27	(1.46)	0.87	0.05	5.49	(1.66)	5.89	(1.55)	−6.59 ***	−0.20	11.60 ***	0.39	3.88 ***	0.25
Positive reframing	5.24	(1.81)	5.31	(1.69)	0.96	0.05	4.76	(1.68)	5.04	(1.63)	−5.33	−0.16	8.19 ***	0.28	2.62 *	0.17
Humor	3.69	(1.72)	3.94	(1.76)	−0.95	−0.05	3.30	(1.51)	3.51	(1.61)	−4.37 ***	−0.13	7.27 ***	0.24	4.09 ***	0.26
Active coping	3.75	(1.51)	4.02	(1.51)	−2.38 *	−0.14	3.81	(1.46)	4.06	(1.49)	−5.04 ***	−0.15	−1.27	−0.04	−0.41	−0.03
Planning	4.49	(1.78)	4.49	(1.75)	1.45	0.08	4.24	(1.63)	4.32	(1.63)	−0.90	−0.03	4.42 ***	0.15	1.60	0.10
Using instrumental support	3.77	(1.68)	4.20	(1.74)	−3.26 **	−0.19	4.03	(1.73)	4.35	(1.76)	−6.16 ***	−0.18	−4.63 ***	−0.16	−1.33	−0.09
Using emotional support	4.03	(1.81)	4.30	(1.72)	−1.46	−0.08	4.54	(1.83)	4.58	(1.79)	0.50	0.02	−8.30 ***	−0.28	−2.45 *	−0.16
Venting	4.23	(1.75)	4.72	(1.81)	−3.12 **	−0.18	4.27	(1.66)	4.53	(1.66)	−4.08 ***	−0.12	−0.70	−0.02	1.68	0.11
Religion	2.83	(1.53)	2.74	(1.35)	0.56	0.03	3.10	(1.75)	2.91	(1.57)	1.05	0.03	−4.83 ***	−0.16	−1.73	−0.11
Denial	3.01	(1.36)	2.81	(1.17)	2.67 **	0.15	3.15	(1.47)	2.92	(1.27)	3.65 ***	0.11	−3.03 **	−0.10	−1.40	−0.09
Self-blame	3.84	(1.70)	3.91	(1.68)	−1.28	−0.07	4.53	(1.86)	4.43	(1.80)	1.51	0.05	−11.44 ***	−0.38	−4.52 ***	−0.29
Self-distraction	4.82	(1.61)	5.00	(1.48)	−2.38 *	−0.14	5.17	(1.53)	5.16	(1.53)	2.87 **	0.09	−6.69 ***	−0.22	−1.57	−0.10
Behavioral disengagement	3.74	(1.58)	3.70	(1.55)	0.11	0.01	4.21	(1.70)	3.83	(1.56)	7.69 ***	0.23	−8.53 ***	−0.29	−1.31	−0.08
Substance use	2.61	(1.37)	2.64	(1.30)	−1.86	−0.11	2.81	(1.56)	2.68	(1.39)	1.00	0.03	−3.88 ***	−0.13	−0.49	−0.03

Note. Intra-year comparisons were made using paired samples *t*-tests, and inter-year comparisons (i.e., T1-T1′ and T2-T2′) using independent samples *t*-tests. * *p* < 0.05. ** *p* < 0.01. *** *p* < 0.001.

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
