# Peer review of "Two Years and Four Time Points: Description of Emotional State and Coping Strategies of French University Students during the COVID-19 Pandemic"

_viruses, 2022, doi:10.3390/v14040782_

Round 1

Reviewer 1 Report

Thank you for the opportunity to review this manuscript.  The objective of this study was to compare the mental health and coping of French university students during the different phases of the pandemic in the first two years.

I think that the theoretical background should include more informations about levels of depression and anxiety in French students also before the pandemic. If the information is available I would like to see more information about type of studies in which students are enrolled. Also, in the discussions the authors stated that the sample was female mostly (p 6 discussion row 37). Do they have any information about the gender distribution of the students in the universities from where the students are?

Also regarding coping strategies a comparison between male/female will be interesting. For ''concern about professional future'' are there any differences between 1st and 3rd year students?

Author Response

Thank you for your invitation to revise our manuscript. You highlighted several issues that needed particular attention. We explain below how we have dealt with these issues. We hope that you will find that we have addressed all your comments. All changes appear in green in the manuscript.

Comment 1: I think that the theoretical background should include more informations about levels of depression and anxiety in French students also before the pandemic.

Data on student depression and anxiety levels, both worldwide and in France, have been added:

“Indeed, even before the pandemic, university students had been identified as having poorer mental health than their nonstudent peers [16–18], with 53% of students reporting having experienced depressive feelings since they started university [19]. A recent systematic review showed a prevalence of 26.1% for depressive symptoms and 24.5% for anxiety among university students [20]. In France, 30% of French university students had symptoms of depression [21], and 5.9% have a major depressive episode [22]. In addition, 55% French university students feel anxious [23], they have major concerns related to academic success, their professional future, time management, exams and financial difficulties [24].”

Comment 2: If the information is available. I would like to see more information about type of studies in which students are enrolled.

The different fields of study of the participants were added to Table 1.

Comment 3: Also, in the discussions the authors stated that the sample was female mostly (p 6 discussion row 37). Do they have any information about the gender distribution of the students in the universities from where the students are?

Although we understand the relevance of this remark, it is unfortunately impossible to answer it.  Indeed, the ratio of men to women varies greatly depending on the faculty (Sciences, Sports Sciences, Psychology, Art, etc.) and as you can see in Table 1, they are very diverse in our sample.

Comment 4: Also regarding coping strategies a comparison between male/female will be interesting. For ''concern about professional future'' are there any differences between 1st and 3rd year students?

The effect of gender on our different variables of interest has been added in Appendix A of the manuscript. The results are quite consistent with the existing literature on the subject, i.e. higher anxiety and depressive symptoms in women (e.g., post hoc tests indicate that women tend to be more anxious than men at T1: t = -7.61, p<.001, d = -.51), and slightly different strategies depending on gender. More precisely, women tend to use more the following coping strategies compared to men: seeking emotional (e.g., at T1: t = -6.57, p<.001, d =-.44) and instrumental support (e.g., at T1: t = -6.4, p<.001, d = -.43), emotion expression (e.g., at T1: -5,27, p<.001, d = -.35), self-blame (e.g., at T1: t = -3.69, p<.001, d = -.25) and denial (e.g., at T1: t = -5.38, p<.001, d = -.36). Men tend to use more acceptance (e.g., at T1: t = 4.21, p<.001, d = -.28) and humor (e.g., at T1: t = 6.35, p<.001, d = 0.42) compared to women. However, these trends are not found in all measurement times.

In the same way, the effect of the level of education on our different variables is extremely weak and is not found in the different times. For example, 3rd year students were found to be more anxious than 1st years, but only for T1. In addition, maladaptive coping strategies seem to be less common in higher study levels. For example, at T1’, PhD students use less frequently denial and self-blame compared to first (t = -3.56, p < .01, d = -0.43), second (t = -3.08, p < .05, d = -0.38) or third year students (t = -2.98, p < .05, d = -0.37).

However, given 1. the very different sample sizes by level of study and gender, and 2. the fact that the effects are not observed across all measurement times, these results do not seem to us to be reliable enough to constitute important findings of this study.

The following sentences have been added to the manuscript

“It is important to note that socio-demographic variables (age, level of education) have little influence on our variables of interest. Only gender seemed to have a small effect on the emotional state and coping strategies of our participants (See Appendix A). More precisely, women have higher levels of anxiety and depressive symptoms than men. Women tend to use more the following coping strategies compared to men: seeking emotional and instrumental support, emotion expression, self-blame and denial. On the contrary, men tend to use more acceptance and humor compared to women. However, these trends are not found in all measurement times.”

-----

We thank for the thorough and relevant comments which, we believe, have helped us make the manuscript stronger. We hope that you will find that these modifications adequately address the suggestions made. Please let us know should you need any further information.

Reviewer 2 Report

Dear authors,

this paper is a novelty in researching impact of the covid pandemic on mental health because it involves longitudinal monitoring/research.

The results are interesting, although partly expected.

In discussion part - line 25 to 35, you gave an explanation why there have been changes in coping styles with stressful situations. However, the question arises as to whether the pandemic itself is uncertain, and the defence mechanisms have been exhausted and brought to a certain point of breaking the ego forces and healthy state. It would be good to give more explanations why the obtained results are exactly like that.

Author Response

Thank you for your invitation to revise our manuscript. You highlighted several issues that needed particular attention. We explain below how we have dealt with these issues. We hope that you will find that we have addressed all your comments. All changes appear in green in the manuscript

Comment 1: The results are interesting, although partly expected.

You are right, the result section has been more developed in the revised version of the manuscript. We hope that these changes will meet your expectations. The following sentences have been added:

“In accordance with our first hypothesis, means comparisons (Table 2) show that students have higher levels (d between .37 and .59) of anxiety and depression during periods of lockdown (T1 and T1’) than during periods after lockdown (T2 and T2’). Comparisons of proportions (Appendix B) show higher proportions of students pre-senting possible anxious and depressive symptoms during the two lockdown periods than during the two period after lockdown. In addition, comparisons of means (Table 2) show that their concerns about the health of their relatives (d between .50 and .81), their health (.30 < d < .39), and their professional future (.11 < d < .25), are greater during pe-riods of lockdown (T1 and T1') than during periods after lockdown (T2 and T2'). Finally, concerning coping strategies, our results indicate a slightly (d < .20) lower use of adapted coping strategies (e.g., active coping, instrumental support), and a slightly (d < .30) higher use of maladaptive coping strategies (e.g., denial, behavioral disengagement) during the lockdown periods compared to the periods after lockdown.

In accordance with our second hypothesis, participants assessed in the second year of the pandemic (2021) have slightly higher levels (.17 < d < .31) of anxiety and depres-sive symptoms, than those interviewed in the first (2020). More precisely, there are higher proportions of students (Appendix B) presenting possible or probable anxious and depressive symptoms during the 2021 lockdown (T1') than during the 2020 lock-down (T1). Comparisons of the two groups of students during the two post-lockdowns periods (T2 vs. T2') highlight slightly higher levels (d = -.28) of concerns about their rel-atives' health in 2021 than in 2020, but no significant differences are observed regarding their concerns about their own health. Furthermore, they are a little more (.17 < d < .31) concerned about their professional future during the 2021 lockdown (T1’) than during the 2020 lockdown (T1), but no significant difference is observed during the post-lockdown periods between the two years (T2 and T2'). In addition, comparisons of the two groups of students during the two lockdowns (T1 vs. T1') highlight slightly lower levels of use of coping strategies in 2021 than in 2020, particularly adapted strat-egies such as acceptance (d = .39) and positive reframing (d = .28). On the other hand, they tend to use a little more instrumental support (d = -.16), and emotional support (d = -.28). In addition, they used a slightly more maladaptive coping strategies such as self-blame (d = -.38), behavioral disengagement (d = -.29), self-distraction (d = -.22), sub-stance use (d = -.13) and denial (d = -.10). Finally, comparisons of the two groups of students during the two periods after lockdowns (T2 vs. T2') show quite similar trends for adaptative strategies but with smaller effect sizes. On the other hand, regarding maladaptive strategies, only self-blame seems to be used more at T2’ than at T2 (d = -.29).”

 Comment 2: In discussion part - line 25 to 35, you gave an explanation why there have been changes in coping styles with stressful situations. However, the question arises as to whether the pandemic itself is uncertain, and the defence mechanisms have been exhausted and brought to a certain point of breaking the ego forces and healthy state. It would be good to give more explanations why the obtained results are exactly like that.

To better explain our results, the following elements have been added.

“This can be explained by the repeated, uncontrollable, and uncertain nature of the stressful events related to the pandemic. Indeed, over time, people realized that there was nothing they could do directly to solve the problem, but that they could simply follow the government's rules (e.g., stay home), which was largely reinforced by information from the media [48]”.

In addition, the importance of implementing interventions that focus on coping strategies was also added.

“All this suggests that university students, who are already known to be a vulnerable population, may be particularly at risk during the 2022 year. It is therefore essential that universities’ preventive medicine departments provide adequate psychological support for students by focusing on interventions to reduce their anxiety and depressive symptoms, and improve their stress adjustment strategies. According to [50], offering practical advice on coping strategies and stress management would help reduce the negative consequences of the COVID-19 pandemic.. ”.

Note: in your evaluation you stated that the introduction “can be improved”. However, you did not comment specifically on this part of the manuscript. We have improved the introduction by adding information about students' anxiety and depressive symptoms prior to the pandemic. We hope that this will meet your expectations.

We thank for the thorough and relevant comments which, we believe, have helped us make the manuscript stronger. We hope that you will find that these modifications adequately address the suggestions made. Please let us know should you need any further information.

Reviewer 3 Report

It was with great pleasure that we read the article entitled "Two years and four time points: comparison of emotional state and adjustment of French university students during the COVID-19 pandemic".
The article judiciously compares two periods in the year 2020 with the year 2021. 
Two conclusions emerge, the first that periods of containment are more impactful than periods of decontainment, the second that the longer the pandemic lasts, the more we observe mental health scores deteriorating. 
The article is well written and of high quality. We believe it can be published as is.

Author Response

Thank you very much for reviewing our article and for giving a favorable recommendation for its publication. 

Reviewer 4 Report

First of all, thank you very much for inviting me to review the article entitled “Two years and four time points: comparison of emotional state and adjustment of French university students during the COVID-19 pandemic.” The paper focuses on university students’ mental health and coping strategies during the during the different phases of the pandemic lockdown.

Abstract – a sentence describing the statistical analysis done would be helpful to the readers. For instance, “Paired samples t-test were……… Our results show that maladaptive....”

Several recommendations/questions for the authors:

During lockdown students are using online learning? Should provide a description on what it means for the students – national lockdown. Since the paper is on the adjustment of French university students, was hoping to determine how to adjust for the better – or enhanced students coping, however, the study was more of a descriptive approach. Maybe minor change of title perhaps.

Methodology and statistical analysis are sound; presentation of results can be expanded. How about the reliability and validity of the data? Table was provided however the findings are not presented, only a few sentence is not enough. This should be explained in details, which is lacking. How about the variables without significant differences? How about the Cohen d’s (effect sizes) these were all not mentioned in the paper, however, are provided within the table. Gender differences? Age? Study level? Demographics are included, however, not considered in the analysis?

Conclusions, implications, and limitations - authors could think about how the investigation ended, and make more specific suggestions for universities. Was thinking on how counselling could be achieve using online facilities in times of lockdown. Can the study be used as a way of looking into who are the most vulnerable or at risk and in need of assistance?

In sum, the study is quite interesting, however, some clarification are needed and additional implications is a must.

Author Response

Thank you for your invitation to revise our manuscript. You highlighted several issues that needed particular attention. We explain below how we have dealt with these issues. We hope that you will find that we have addressed all your comments. All changes appear in green in the manuscript

Comment 1: Abstract – a sentence describing the statistical analysis done would be helpful to the readers. For instance, “Paired samples t-test were……… Our results show that maladaptive....”

A sentence describing the statistical analyses has been added to the abstract:

“In order to compare scores between the lockdown and deconfinement periods within the same year paired samples t-test have been performed. To compare scores between our two years including different participants, independent samples t-test were conducted.”

Comment 2: During lockdown students are using online learning? Should provide a description on what it means for the students – national lockdown.

You are right. In France, during the lockdown, all the lessons were done online. During periods without lockdown but with high virus circulation, some lessons were conducted online and others face-to-face. These elements have been added to the manuscript.

“Data were collected anonymously at four timepoints, via an online survey: (1) during France’s first national lockdown (23 April-10 May 2020; nT1 = 1357; Mage = 21.22 years ± 4.64) during which all lessons were conducted online.; (2) during the period after lockdown (9‑26 June 2020; nT2 = 309; Mage = 22.24 years ± 5.93) with some of the evaluations taking place online, and some face-to-face; (3) 1 year after the first lockdown, which was also a lockdown period (23 April-10 May 2021; nT1’ = 2569; Mage = 21.45 years ± 4.19) during which all lessons were conducted online; and (4) 1 year after the first deconfinement (9‑26 June 2021, nT2’ = 1136; Mage = 21.63 years ± 4.58) with some of the evaluations taking place online, and some face-to-face.”

Comment 3: Since the paper is on the adjustment of French university students, was hoping to determine how to adjust for the better – or enhanced students coping, however, the study was more of a descriptive approach. Maybe minor change of title perhaps.

You are absolutely right, our study is essentially descriptive and aims to compare the strategies employed at different times. As per your advice, the title has been changed to

“Two years and four time points: description of emotional state and coping strategies of French university students during the COVID-19 pandemic”

Comment 4: Methodology and statistical analysis are sound; presentation of results can be expanded. How about the reliability and validity of the data? Table was provided however the findings are not presented, only a few sentence is not enough. This should be explained in details, which is lacking.

Thank you for your positive feedback. In accordance with your advice, the result section has been more developed in the revised version of the manuscript. We hope that these changes will meet your expectations.

“In accordance with our first hypothesis, means comparisons (Table 2) show that students have higher levels (d between .37 and .59) of anxiety and depression during periods of lockdown (T1 and T1’) than during periods after lockdown (T2 and T2’). Comparisons of proportions (Appendix B) show higher proportions of students presenting possible anxious and depressive symptoms during the two lockdown periods than during the two period after lockdown. In addition, comparisons of means (Table 2) show that their concerns about the health of their relatives (d between .50 and .81), their health (.30 < d < .39), and their professional future (.11 < d < .25), are greater during periods of lockdown (T1 and T1') than during periods after lockdown (T2 and T2'). Finally, concerning coping strategies, our results indicate a slightly (d < .20) lower use of adapted coping strategies (e.g., active coping, instrumental support), and a slightly (d < .30) higher use of maladaptive coping strategies (e.g., denial, behavioral disengagement) during the lockdown periods compared to the periods after lockdown.

In accordance with our second hypothesis, participants assessed in the second year of the pandemic (2021) have slightly higher levels (.17 < d < .31) of anxiety and depressive symptoms, than those interviewed in the first (2020). More precisely, there are higher proportions of students (Appendix B) presenting possible or probable anxious and depressive symptoms during the 2021 lockdown (T1') than during the 2020 lockdown (T1). Comparisons of the two groups of students during the two post-lockdowns periods (T2 vs. T2') highlight slightly higher levels (d = -.28) of concerns about their relatives' health in 2021 than in 2020, but no significant differences are observed regarding their concerns about their own health. Furthermore, they are a little more (.17 < d < .31) concerned about their professional future during the 2021 lockdown (T1’) than during the 2020 lockdown (T1), but no significant difference is observed during the post-lockdown periods between the two years (T2 and T2'). In addition, comparisons of the two groups of students during the two lockdowns (T1 vs. T1') highlight slightly lower levels of use of coping strategies in 2021 than in 2020, particularly adapted strategies such as acceptance (d = .39) and positive reframing (d = .28). On the other hand, they tend to use a little more instrumental support (d = -.16), and emotional support (d = -.28). In addition, they used a slightly more maladaptive coping strategies such as self-blame (d = -.38), behavioral disengagement (d = -.29), self-distraction (d = -.22), substance use (d = -.13) and denial (d = -.10). Finally, comparisons of the two groups of students during the two periods after lockdowns (T2 vs. T2') show quite similar trends for adaptative strategies but with smaller effect sizes. On the other hand, regarding maladaptive strategies, only self-blame seems to be used more at T2’ than at T2 (d = -.29).”

Comment 5: How about the variables without significant differences? How about the Cohen d’s (effect sizes) these were all not mentioned in the paper, however, are provided within the table.

As mentioned in response to the previous comment, the result part has been further developed. In addition, the statistical analysis section has been completed.

“Effect sizes are expressed with Cohen’s d. Data were analyzed using the JASP software (version 0.11.1).”

Comment 6: Gender differences? Age? Study level? Demographics are included, however, not considered in the analysis?

You are right. Considering 1. the heterogeneity of our participants for these different variables and 2. The presence of different measurement times in our study, this information seems difficult to exploit for this study. These data are reported in the manuscript primarily to describe our sample, but our hypotheses do not consider this information, which justifies not exploiting it in our analyses. However, we have carried out these different analyses in order to respond more precisely to your comment.

The effect of gender on our different variables of interest has been added in Appendix A of the manuscript. The results are quite consistent with the existing literature on the subject, i.e. higher anxiety and depressive symptoms in women (e.g., post hoc tests indicate that women tend to be more anxious than men at T1: t = -7.61, p<.001, d = -.51), and slightly different strategies depending on gender. More precisely, women tend to use more the following coping strategies compared to men: seeking emotional (e.g., at T1: t = -6.57, p<.001, d =-.44) and instrumental support (e.g., at T1: t = -6.4, p<.001, d = -.43), emotion expression (e.g., at T1: -5,27, p<.001, d = -.35), self-blame (e.g., at T1: t = -3.69, p<.001, d = -.25) and denial (e.g., at T1: t = -5.38, p<.001, d = -.36). Men tend to use more acceptance (e.g., at T1: t = 4.21, p<.001, d = -.28) and humor (e.g., at T1: t = 6.35, p<.001, d = 0.42) compared to women. However, these trends are not found in all measurement times.

In the same way, the effect of the level of education on our different variables is extremely weak and is not found in the different times. For example, 3rd year students were found to be more anxious than 1st years, but only for T1. In addition, maladaptive coping strategies seem to be less common in higher study levels. For example, at T1’, PhD students use less frequently denial and self-blame compared to first (t = -3.56, p < .01, d = -0.43), second (t = -3.08, p < .05, d = -0.38) or third year students (t = -2.98, p < .05, d = -0.37).

However, given 1. the very different sample sizes by level of study and gender, and 2. the fact that the effects are not observed across all measurement times, these results do not seem to us to be reliable enough to constitute important findings of this study.

Finally, regarding age, the more students are old, the more they tend to use certain coping strategies. For example, at T1, the more students are old, the more they tend to use humor to cope with stress (r = .08, p<.05); at T2’ the more students are old, the more they tend to use substance to cope with stress (r = .07, p<.05). However, here again, these associations are not found over all measurement times, and the effects, even if significant, are extremely weak (as an indication, the two correlations presented as examples are the highest).

The following sentences have been added to the manuscript

“It is important to note that socio-demographic variables (age, level of education) have little influence on our variables of interest. Only gender seemed to have a small effect on the emotional state and coping strategies of our participants (See Appendix A). More precisely, women have higher levels of anxiety and depressive symptoms than men. Women tend to use more the following coping strategies compared to men: seeking emotional and instrumental support, emotion expression, self-blame and denial. On the contrary, men tend to use more acceptance and humor compared to women. However, these trends are not found in all measurement times.”

Comment 7: Conclusions, implications, and limitations - authors could think about how the investigation ended, and make more specific suggestions for universities. Was thinking on how counselling could be achieve using online facilities in times of lockdown. Can the study be used as a way of looking into who are the most vulnerable or at risk and in need of assistance?

Our study is primarily descriptive, so it seems complicated to use our results to make very specific recommendations and/or to identify the most vulnerable students. However, our results show that student vulnerability is increased during confinement and in the second year of the pandemic. Based on this, we made the following general recommendations in the discussion:

“All this suggests that university students, who are already known to be a vulnerable population, may be particularly at risk during the 2022 year. It is therefore essential that universities’ preventive medicine departments provide adequate psychological support for students by focusing on interventions to reduce their anxiety and depressive symptoms, and improve their stress adjustment strategies. According to [50], offering practical advice on coping strategies and stress management would help reduce the negative consequences of the COVID-19 pandemic. Furthermore, since periods of lockdowns are the riskiest, it seems essential to be able to implement therapeutic strategies that limit contact. To do so, online self-help interventions (e.g., web page, Facebook page, e-learning) seem to be promising tools. It was identified as being particularly valuable for university students [51–53] and has proven to be effective for students during the pandemic [54].”

And we have added the following in the conclusion:

“This supports the hypothesis that the psychological effects of COVID-19 may persist even after the pandemic and highlights the need to continue to deploy interventions (face-to-face and/or online) aimed at reducing students' depressive and anxiety symptoms, as well as improving their stress adjustment strategies.”

In sum, the study is quite interesting, however, some clarification are needed and additional implications is a must.

We thank for the thorough and relevant comments which, we believe, have helped us make the manuscript stronger. We hope that you will find that these modifications adequately address the suggestions made. Please let us know should you need any further information.

Comment 7: Conclusions, implications, and limitations - authors could think about how the investigation ended, and make more specific suggestions for universities. Was thinking on how counselling could be achieve using online facilities in times of lockdown. Can the study be used as a way of looking into who are the most vulnerable or at risk and in need of assistance?

Our study is primarily descriptive, so it seems complicated to use our results to make very specific recommendations and/or to identify the most vulnerable students. However, our results show that student vulnerability is increased during confinement and in the second year of the pandemic. Based on this, we made the following general recommendations in the discussion:

“All this suggests that university students, who are already known to be a vulnerable population, may be particularly at risk during the 2022 year. It is therefore essential that universities’ preventive medicine departments provide adequate psychological support for students by focusing on interventions to reduce their anxiety and depressive symptoms, and improve their stress adjustment strategies. According to [50], offering practical advice on coping strategies and stress management would help reduce the negative consequences of the COVID-19 pandemic. Furthermore, since periods of lockdowns are the riskiest, it seems essential to be able to implement therapeutic strategies that limit contact. To do so, online self-help interventions (e.g., web page, Facebook page, e-learning) seem to be promising tools. It was identified as being particularly valuable for university students [51–53] and has proven to be effective for students during the pandemic [54].”

And we have added the following in the conclusion:

“This supports the hypothesis that the psychological effects of COVID-19 may persist even after the pandemic and highlights the need to continue to deploy interventions (face-to-face and/or online) aimed at reducing students' depressive and anxiety symptoms, as well as improving their stress adjustment strategies.”

In sum, the study is quite interesting, however, some clarification are needed and additional implications is a must.

We thank for the thorough and relevant comments which, we believe, have helped us make the manuscript stronger. We hope that you will find that these modifications adequately address the suggestions made. Please let us know should you need any further information.

Round 2

Reviewer 4 Report

First of all, thank you so much for inviting me to review the revised version of the paper. As per going over the authors' reply and the reflected revisions provided in the paper. Evidence of improvement are seen. The current version is now much better and clearer. 

Just one simple question: Did you compute for the survey reliability? Cronbach Alpha? was not able to find it